# Machine learning for cluster analysis of localization microscopy data

David J. Williamson [1], Garth L. Burn[1], Sabrina Simoncelli[1,2], Juliette Griffié[1], Ruby Peters[1], Daniel M. Davis [3] & Dylan M. Owen [1,4✉]

Quantifying the extent to which points are clustered in single-molecule localization microscopy data is vital to understanding the spatial relationships between molecules in the underlying sample. Many existing computational approaches are limited in their ability to process large-scale data sets, to deal effectively with sample heterogeneity, or require subjective user-defined analysis parameters. Here, we develop a supervised machine-learning approach to cluster analysis which is fast and accurate. Trained on a variety of simulated clustered data, the neural network can classify millions of points from a typical single-molecule localization microscopy data set, with the potential to include additional classifiers to describe different subtypes of clusters. The output can be further refined for the measurement of cluster area, shape, and point-density. We demonstrate this approach on simulated data and experimental data of the kinase Csk and the adaptor PAG in primary human T cell immunological synapses.

[1] Department of Physics and Randall Centre for Cell and Molecular Biophysics, King's College London, London, UK. [2] London Centre for Nanotechnology and Department of Chemistry, University College London, London, WC1H 0AH, UK. [3] Division of Infection, Immunity and Respiratory Medicine, University of Manchester, Manchester, UK. [4] Institute of Immunology and Immunotherapy, Department of Mathematics and Centre for Membrane Proteins and Receptors, University of Birmingham, Birmingham, UK. ✉email: d.owen@bham.ac.uk

O nce raw single-molecule localization microscopy (SMLM) data have been obtained and the positions of emitters localized, there is the further challenge of how to effectively analyze the resulting data. Whereas images from wide-field or confocal microscopes are composed of arrays of pixels, the data from SMLM[1,2] are fundamentally a list of coordinates, each representing the spatial location of a point-emitter. This list can be plotted and rasterized for examination with conventional image analysis tools, but an ideal method would operate on the original coordinate data without requiring its transformation. A distribution of points within the imaging field can be interrogated using spatial point pattern analyses to reveal the spatial relationships between the points and higher-scale relationships between clusters of points, or between points from different imaging channels. Such techniques include Ripley's $K$ Function[3], and a local generalization as Getis & Franklin's local point pattern analysis[4], as well as the radial distribution (or pair correlation) functions[5], and DBSCAN[6].

Common among many of these approaches is the selection of analysis parameters, which can lead to a suboptimal interpretation of the data, for example, when points are clustered at a different spatial scale to the one used for assessment or when points are not homogeneously clustered. These situations are common in the case of complex data from biological specimens. Often these methods also require a threshold setting in order to label points as 'clustered' or 'not clustered'. As the clustering values can depend strongly on the overall density and arrangement of points, it is likely that the appropriate threshold for one image will be unsuitable for the next. A solution can be found in model-based cluster analysis, such as Bayesian inference[7], where cluster analysis outputs are scored against a model of clustering, allowing the best-scoring set of analysis parameters to be selected. Although this method removes the problem of selecting the "best" parameters, it is computationally intensive and therefore not practical for large data sets, typically requiring the use of cropped regions-of-interest selected from each image[8].

Machine learning is a computational and statistical approach to extract meaningful information from complex data where a fully descriptive model is not otherwise available. Machine learning has developed rapidly within the field of artificial intelligence, where it has been predominantly employed in the service of problems such as facial recognition, autonomous vehicle navigation, and speech recognition. Two principal approaches to conducting machine learning are either as supervised or unsupervised. Supervised learning requires examples of known labels or patterns, which are then used to extract similar patterns from unfamiliar data. In unsupervised learning, data sets are used to discover new patterns and labels, without any prior information. One of the tools used in machine learning is that of neural networks where 'neurons' are self-contained units capable of accepting input, processing that input, and generating an output. Neurons performing similar functions can be grouped together as layers, and layers can be connected to form the neural network. The machine can adjust the internal parameters of the layers in order to optimize the network's performance in translating the given input into the desired output.

Machine learning, when applied to microscopy images, often uses convolutional neural networks to examine raster-based images. Such networks were originally developed and refined to solve computer vision problems and have proven useful in the identification and extraction of abstract features from images. However, localization microscopy data are not immediately compatible with convolutional neural networks. Here, we employ machine learning to solve realistic cluster identification problems in SMLM data sets by training a neural network to extract features from nearest-neighbor distance-derived data, where such

data might be acquired by (F-)PALM[1,9], (d)STORM[2,10], GSD (IM)[11,12], or (DNA-)PAINT[13,14] techniques. Note, however, that the network does not correct for artifacts inherent to the sample preparation or imaging of these individual SMLM methods (such as fluorophore re-blinking and sample drift). These artifacts should be corrected for beforehand, or the results interpreted in their context.

A set of software modules is presented to prepare raw data, train new models, evaluate data with trained modules, and describe cluster properties. A quantitative comparison with other methods demonstrates the accuracy of the approach, with benefits of being able to access arbitrary regions-of-interest from very large data sets and rapid processing time.

## Results

**Model specification, implementation and workflow.** The models described here operate on each point from a data set consisting of a list of $x$ and $y$ coordinates, in turn. Each model's input is an array of values derived from each point's nearest-neighbor distances and its output is a binary label, indicating whether the point has been classified as either 'not clustered' or 'clustered'.

For a point within a spatial point-pattern, a sequence can be constructed of the distances from that point to its neighbors. The monotonic sequence of near-neighbor distances (or the difference in distances between consecutive near neighbors) are here used as input for machine learning models. Models were constructed using Keras[15], an open-source machine-learning framework for Python, which assembles neural network layers in a linear sequence. A complete model comprises the sequence of processing layers and their various internal values. These internal values are adjusted during the training process to maximize the accuracy of the model by comparing the model's final output with the expected output and, over many training iterations, the model's configuration converges to its most capable configuration.

The workflow, with specific Python scripts used at different stages, is shown in Fig. 1. To begin training a new model, data with known clustering characteristics are simulated and prepared by measuring the distances for each point to its N nearest neighbors. Next, a model configuration is specified in Keras as a sequential stack of layers; each performing a specific processing task on the input. Three example models are described here; models are given a unique six-character name to help identify and differentiate between them.

One model, designated 'XPILJZ', consists of an input layer for 100 near-neighbor values followed by two fully connected layers and an output layer indicating if the input values describe a point that is clustered or not clustered. This model presents a very simple arrangement of layers for feature extraction and classification.

Another model, 07VEJJ, also uses 100 near neighbor input values but contains additional layers: a one-dimensional convolutional layer is employed to exploit any existing correlation between the near neighbor distances. Max pooling layers are employed to extract prominent features from the data (such as a large change in near neighbor distance) and dropout layers are used to reduce overfitting of the model to the training data. In addition, two long short-term memory (LSTM) layers are used, as they were expected to increase classification accuracy owing to the sequential nature of the input data[16]. An LSTM is a type of recurrent neural network formed from a chain of network units, which allows information to persist (as 'memory') and allows the network to learn long-term dependencies from the input sequence. Furthermore, stacking LSTM layers (as in model 07VEJJ) may permit a deeper abstraction of the input sequence and help boost the model's accuracy[17].

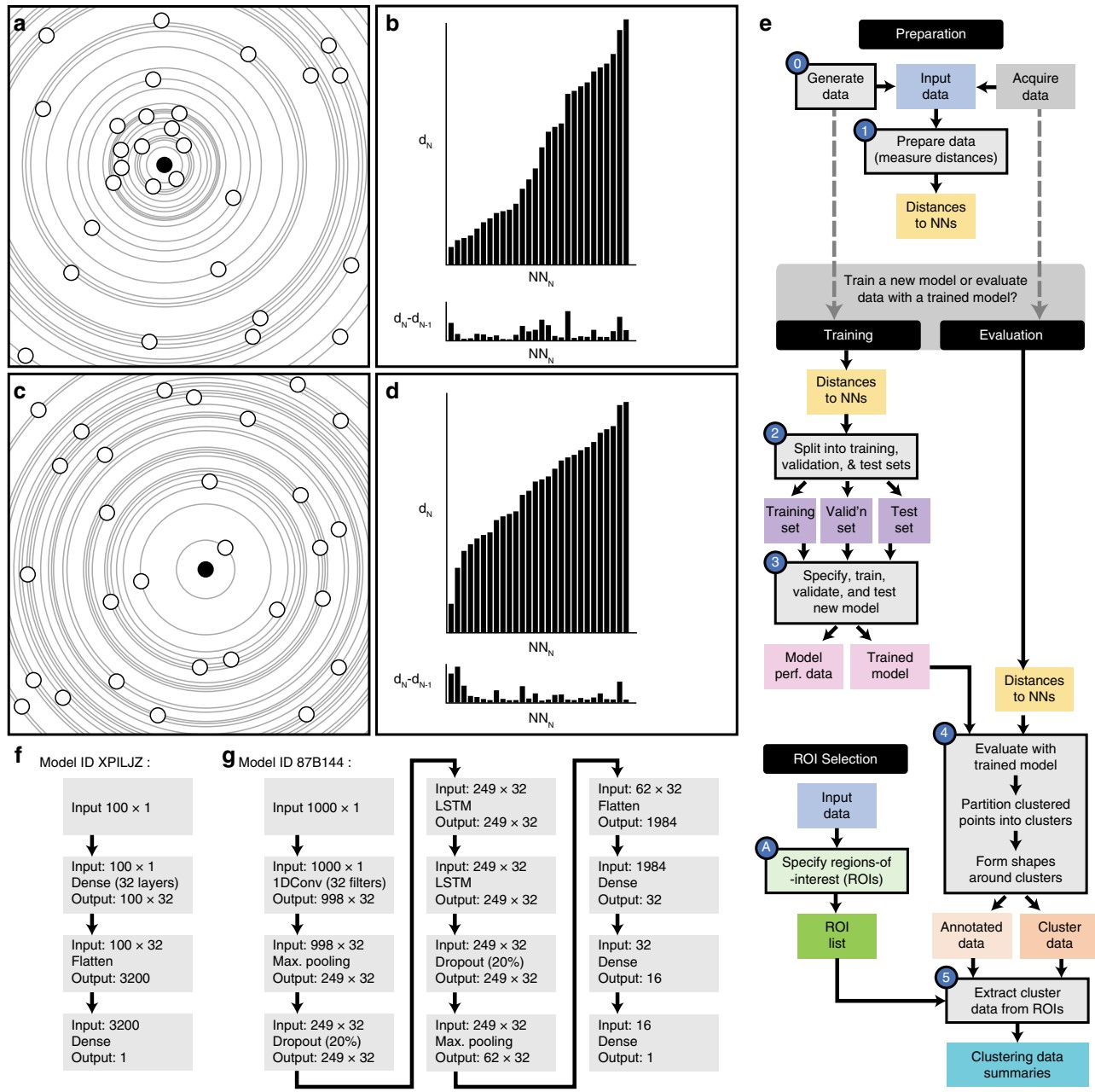

**Fig. 1 Method workflow. a** A collection of points in a clustered configuration with a reference point highlighted in solid black. **b** The distances from the reference point to its consecutive neighbors (top) and the difference between consecutive distances (bottom). **c, d** Points in a non-clustered arrangement and distances for a single highlighted point. **e** Workflow; colored shapes indicate input or output files. Shapes with black border and a number are stages handled by software described here. **f, g** show neural network layer configurations for models XPILJZ and 87B144.

The number of near neighbors used for input should exceed the anticipated number of points in the largest cluster (by point count) in the data. For this reason, we created a third model, 87B144, with the same layer configuration as 07VEJJ except for the input layer, which requires 1000 near neighbor distances as input. This model is expected to be able to identify points in data sets featuring larger, denser clusters but at the expense of increased computational and memory costs.

Simulated data sets were generated to resemble experimental clustered distributions with the distribution of points restricted within a 'cell-like' shape, mimicking a T-cell synapse formed against a flat surface and imaged by total internal reflection fluorescence (TIRF) microscopy[18,19]. This was done to represent the gross morphology seen in a typical cell synapse, such as

membrane protrusions with uneven edges, and other constrained geometries. Within the cell-like shape, points are then distributed according to a clustering scenario, for example: '100 points per µm², 50% of points in clusters, 10 points per cluster, and clustered points are located up to 30 nm from a cluster center'. Ranges of cluster descriptors in combination can therefore yield multifarious cluster scenarios. This method of cluster generation was chosen for its simplicity and that it can generate a wide variety of data sets containing well defined clusters for which the ground truth of every point is known.

All simulated data sets were prepared for training by determining the distances to the chosen number of nearest neighbors (100 or 1000) for all points in all images. The sequence of distances for all points were pooled and a mixture of input

sequences representing both clustered and non-clustered points were randomly selected and split into sets for training, validation, and testing of new models. The classification labels matching the selected points were similarly extracted and split into three sets. For evaluation of data, the assigned scores and classification labels (clustered, not clustered) for each data point were used to segment the points into like-clusters (see Methods). In brief, each clustered point was grouped into a cluster with its nearest neighbor if that neighboring point was also labeled as clustered. Consecutive neighbors were placed into this growing cluster until the first neighboring non-clustered point was encountered. Cluster shapes were created by centering a disc over each point and taking the union of all shapes; the radius of the disk being proportional to mean nearest-neighbor distance. The resulting outline shape was eroded, proportional to the mean nearest-neighbor distance to form the final cluster shape (see Supplementary Methods and Supplementary Fig. 1).

**Performance on simulated data.** Simulated scenarios for training all three models were compiled from combinations of different, ranged, clustering parameters, specifically: an overall point density, a point population within each cluster, a proportion of the points that are in clusters, and a maximum distance from the cluster seed (effectively the 'cluster radius'). The ranges of each parameter used for training and performance testing are given in Supplementary Note 1. Each clustering parameter combination therefore describes a potential 'clustering scenario'. All training clustering scenario combinations were filtered to create a set of 711 'viable' scenarios, which would yield 1–5 cluster(s) per μm² and a point density inside clusters of between 1.5× and 100× that of the non-clustered point density; point density inside of clusters was determined by dividing the number of points per cluster by the area per cluster. The point density outside was determined by dividing the total number of non-clustered points by the non-clustered area. The non-clustered area was calculated by subtracting the total area covered by all the clusters from the total area of the 'cell' into which points were distributed. Data sets with clusters having density ratios below 1.5× that of the non-clustered area were excluded because they are not readily distinguishable from the non-clustered area or would form 'holes' devoid of points (see Supplementary Table 1). These clustering scenarios were specified based upon the clustering of membrane proteins observed in T cells[18–21]. Clustering in the highest density scenarios is estimated from an expected upper limit of proteins in the plasma membrane. For example, assuming a hydrodynamic radius of 2–3 nm would result in a maximum packing density of ~50,000 proteins per μm². For larger globular proteins with a hydrodynamic radius of 8 nm, the packing density decreases to 4000 per μm². At 25% occupancy of proteins in the membrane, this gives a maximum upper bound in the region of 1000–12,000 proteins per μm². Therefore, our suite of cluster scenarios is expected to include a realistic range of particle densities as well as some more challenging scenarios with very high point densities.

Training data files were then used as input for the pre-processing stage where the nearest-neighbor distances were recorded. Model 07VEJJ was configured with 12 layers in Keras and trained on 500,000 input samples comprising an even mix of clustered and non-clustered labels; a different set of 100,000 input samples were used for model validation during training. The trained model was tested on another set of 100,000 input samples with the same even mix of classification labels. This model demonstrated an accuracy of 92.4% on both training and testing data, with an F1 score (a measure of binary classification accuracy) of 0.9243 for the testing data (precision 0.9245 and recall 0.9243). Ten-fold cross validation of the model resulted in

an accuracy score of 91.8 ± 0.3%. Other models used in this study achieved similar performance: Model 87B144 (12 layers using 1000 near neighbors) showed an accuracy of 94.0% on both training and testing data with an F1 score of 0.9398 for the testing data (precision 0.9420 and recall 0.9399). Model XPILJZ (four layers using 100 near neighbors) showed an accuracy of 91.9% on the training data and 92.0% on testing data with an F1 score of 0.9199 for the testing data (precision 0.9204 and recall 0.9199) and was able to recover information about clusters comparable to the cluster scenarios used to generate test samples (Fig. 2).

The model was then used to evaluate novel simulated data, including images only containing spatially random (non-clustered) points (Fig. 3a, d–g). These data featured a similar range of clustering scenarios as that which was used to derive the training, validation, and testing data sets as well as some scenarios never experienced during training. Specifically, these novel clustering scenarios covered 5, and 10–100 (in steps of 10) percent points clustered, 5 and 10–100 (in steps of 10) nm maximum distance from a cluster center. These parameters were combined with fixed pairs of points per cluster and overall point density parameters, namely 50 points per μm² and 10 points per cluster, 100 points per μm² and 20 points per cluster, 300 points per μm² and 100 points per cluster, and 500 points per μm² and 80 points per cluster. Model 07VEJJ showed >90% accuracy in scenarios without any clustering (only randomly distributed points) where the overall point density was 100 points per μm² or greater. For lower-density data the accuracy decreased to 89.2% in images with 50 points per μm² and 84.5% for those with 10 points per μm². In simulated data sets with clustering present, 07VEJJ was able to correctly identify labels for over 90% of points in most clustering scenarios (Fig. 3d, f). When presented with data outside of the original training data clustering scenarios, for example, clusters containing 100 points, the model was still able to perform the correct classification in ~90% of the cases (Fig. 3e). A qualitative assessment of performance was also performed on a simulated data set containing clusters in lines and rings within a highly variable background of non-clustered points (Supplementary Fig. 2) which demonstrates the models are capable of finding clustered points from a variety of structures in addition to the circular clusters on which the models were trained. Cluster scenarios in which points were distributed within clusters according to a Gaussian function were also tested. Models 07VEJJ and 87B144 both returned accurate classification of points in many scenarios with a decrease in performance on scenarios at the extremes of the parameter ranges (Supplementary Fig. 3). Interestingly 07VEJJ outperformed 87B144, even though it accepts input from fewer neighbors. This could indicate over-fitting of 87B144 to features found in the hard-edged clusters on which it was trained. As our models accept input data from a predefined number of nearby points, we also tested the effect of exceeding this number. Cluster scenarios that presented clusters containing 150 or 200 points per cluster were evaluated with Model 07VEJJ (Supplementary Fig. 4). This model is limited to receiving information from the 100 nearest neighbors when deciding a point's classification and expectedly showed decreased classification performance on highly populated clusters, with these clusters becoming fragmented or ignored.

The performance was compared with Getis & Franklin's local point pattern analysis (G&F LPPA), Bayesian cluster analysis (Fig. 3a–c, g), DBSCAN (Supplementary Fig. 5), and SR-Tesseler[22] (Supplementary Figs. 6 and 7), which have also been used for SMLM data analysis. Where possible, start-to-finish timings to assess computational load were obtained for each method (Supplementary Fig. 8). SR-Tesseler is an interactive application and so any timings would be user-dependent. However, once processing settings had been determined, this

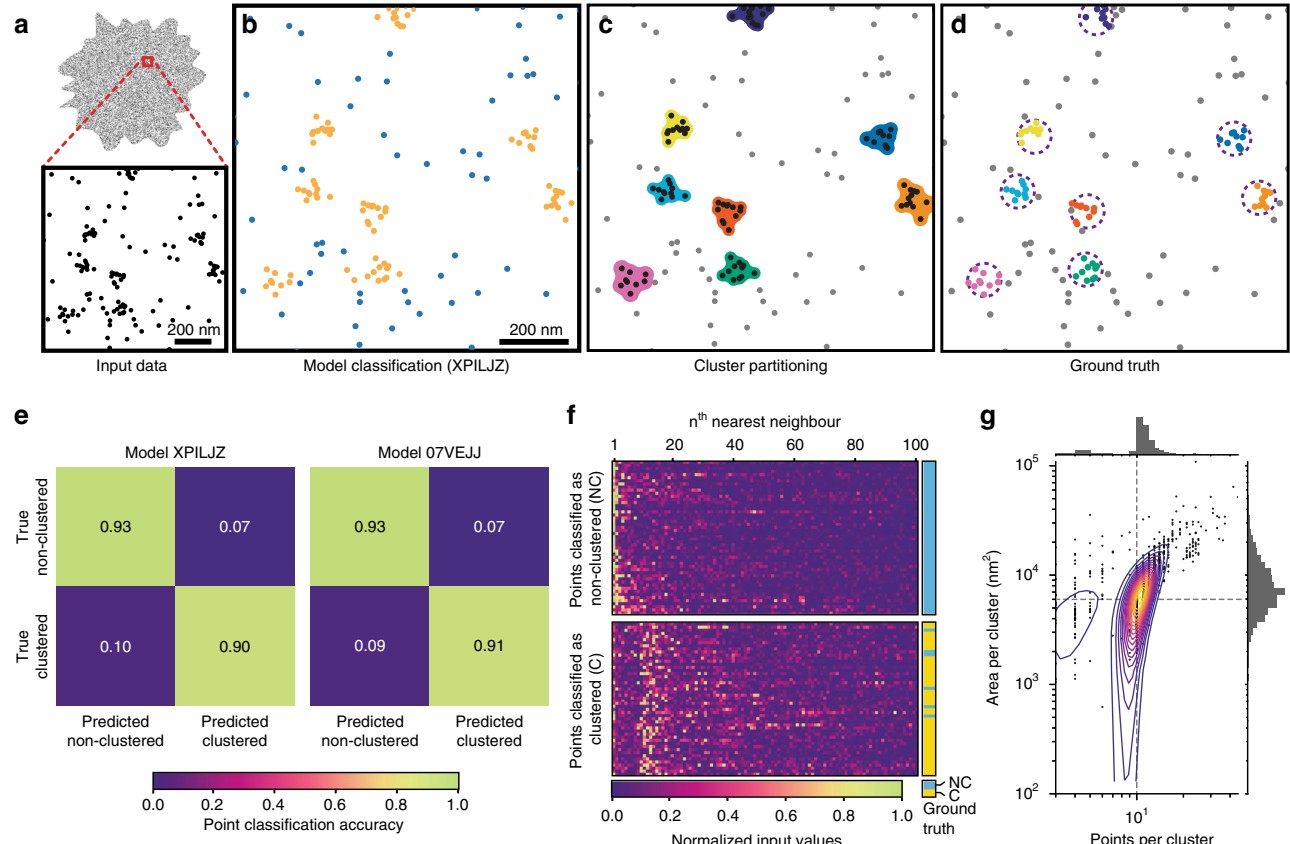

**Fig. 2 Testing of two example models using simulated data. a** Input data are the point coordinates, which are evaluated (**b**) by a model and classified as either clustered (gold) or non-clustered (blue). This information is used to partition the clustered points into spatially similar clusters, around which a cluster shape can be formed (**c**). The original (known) clustering assignment (**d**, dashed circles indicate the maximum distance clustered points may be positioned from the cluster center) may be used in comparison to assess the accuracy of the process. Accuracy is represented as confusion matrices (**e**) for two models generated using different configurations which report good accuracy. **f** A random sample of the normalized input data (to 100 near neighbors), separated into those points which were classified by Model XPILJZ as non-clustered (top) or clustered (bottom); the true label for each point is shown on the right (blue: non-clustered, yellow: clustered). **g** Output data faithfully describes the original clustering scenario, here the clusters contained 10 points within a maximum radius of 40 nm (indicated by dashed lines).

software could complete its analysis of an image within 1–5 min. CAML was slower than DBSCAN for smaller data sets but became as fast or faster when images approached half a million points. All methods were faster than Bayesian analysis, most likely owing to the exhaustive parameter scanning employed by this method.

DBSCAN was quite sensitive to the value of the 'epsilon' parameter, which decides if nearby points are 'neighbors'. Values that worked well for some cluster scenarios were unsuitable for others. In an experimental scenario, this would require careful selection of processing settings for each image. Processing with SR-Tesseler was very fast for the computational stages (seconds to minutes) but also required significant user interaction to conduct a full analysis. Selected clustering scenarios were processed with SR-Tesseler and the clustering statistics were compared (Supplementary Fig. 6). In general, CAML and SR-Tesseler returned similar results for the size and point complement of clusters they identified with CAML often returning a more homogenous set of clusters with properties closer to the target values.

Models are configured with a set input size, which is the number of near-neighbor values to use. Models are also trained on simulated data that contain clusters that are a subset of all possible ways that points may form clusters within a space. As such, the size of a model's input window and the maximum number of points in a cluster seen during training may affect the performance of a model. To test this, various models were created with different input sizes (accepting 50, 100, or 200 near-neighbor values) and then trained on data where clusters could contain up this number of points. Trained models were then tested on data with clusters that were outside of the model's training experience. As anticipated, models performed poorly when the size (in points) of a cluster exceeded the model's input window (Supplementary Figs. 9 and 10). Furthermore, models that were trained on small clusters could find points within much larger clusters, provided they had a sufficiently large input window.

Our models utilize the nearest-neighbor distance as input. To explore whether this approach was necessary, we also trained models supplied with only normalized $xy$ coordinate data. The overall layer specification was maintained, however instead of a one-dimensional input vector of Euclidean distances for a point's 100 or 1000 nearest neighbors, a two-dimensional vector of the neighboring points' $x$ and $y$ coordinates (normalized and relative to the origin point's coordinates) was used. These models were able to achieve similar performance (~94% accuracy) to those trained on Euclidean distances but at the expense of considerably longer training times; for 1000 neighbors the training time, when using normalized coordinates as input, was ~15 days (excluding data preparation time) compared with a few hours for models trained using Euclidean distances (Supplementary Note 2). It is possible that this time can be improved (as for any such training task) through strategies such as early stopping to conclude training once a pre-set performance target is reached. This alternative approach

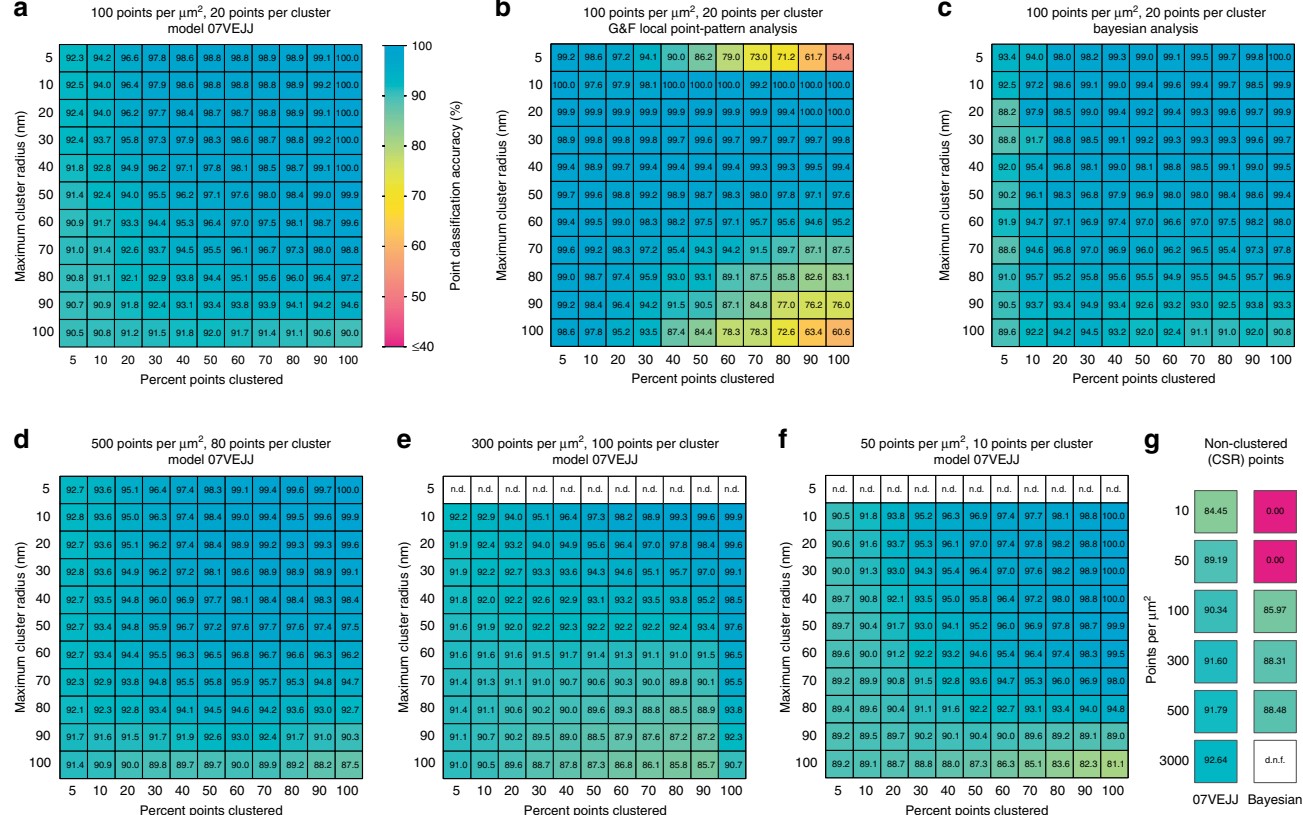

**Fig. 3 Classification accuracy of simulated data.** Data were assessed with **a** model 07VEJJ, or **b** Getis & Franklin's Local Point Pattern Analysis, or **c** Bayesian Cluster Analysis on 3 × 3 μm centered regions extracted from the data sets analyzed by **a** and **b**. Model 07VEJJ shows comparable performance at very high point densities (**d**), when the number of points per cluster matches the limit of the models near-neighbor distances 'reach' (**e**) (n.d. = not determined), and very low point densities (**f**). **g** Classification accuracy for completely spatially random data at different point densities analyzed by Model 07VEJJ or the Bayesian method; d.n.f. = did not finish. Data are mean values from ten replicates.

still required calculation of distance matrices to identify which neighboring points' coordinates to use for training but, as it was neither faster nor more accurate than models trained on the distance values, the Euclidean distance trained models were used for the evaluation of experimental data. Computation times may be further reduced by selecting and processing candidate points for the training pool during the data simulation stage.

**Demonstration on experimental data.** C-terminal Src kinase (Csk) is a protein tyrosine kinase and a well-established negative regulator of T-cell receptor signaling through its inactivation of membrane-associated Src kinases, including the TCR-associated kinase Lck[23]. As a cytosolic protein, Csk is thought to be regulated through its association with the transmembrane adapter protein PAG (phosphoprotein associated with glycosphingolipid-enriched microdomains). In non-activated T cells, PAG is predominantly phosphorylated which facilitates Csk binding[24,25]. Upon TCR stimulation PAG is dephosphorylated and releases Csk[26,27]. Once dissociated from PAG, Csk is removed from the plasma membrane through interactions with adaptor protein TRAF3 (TNF receptor-associated factor 3) and PTNP22 (protein tyrosine phosphatase N22)[28], leading to a more-permissive environment for Src kinase activation. Immune synapses were formed between primary human T cells and an activating antibody-coated glass surface, fixed, stained, and imaged by dSTORM microscopy in a TIRF configuration. Machine learning based cluster analysis using Model 87B144 demonstrated changes in the clustering of Csk and PAG at the plasma membrane (Fig. 4). These changes were dependent on

both the status of the T cells (naive or pre-stimulated) and their activation status (non-activated or activated on anti-CD3 + ICAM-1-coated glass).

Naive cells showed a change in Csk clustering (Supplementary Figs. 11a–f and 12a–d) with activation through an increase in the number of clusters from 4.85 clusters per μm² (median, IQR 3.15–7.38 clusters per μm²) to 8.80 clusters per μm² (median, IQR 6.75–10.75 clusters per μm²), $P < 0.0001$. Pre-stimulated cells after activation showed an increase in the number of clusters from 6.77 clusters per μm² (median, IQR 5.13–8.37 clusters per μm²) to 14.37 clusters per μm² (median, IQR 11.03–18.57 clusters per μm²), $P < 0.0001$. Naive cells also showed an increase in the percentage of points clustering from 53.2% (median, IQR 48.3–59.7%) to 61.1% (median, IQR 53.1–64.4%) for the stimulated condition, $P < 0.0001$ and in the number of points per cluster from 9 (median, IQR 6–14) points per cluster to 11 (median, IQR 7–18) ($P < 0.0001$). The number of points per cluster was also higher in pre-stimulated cells compared with naive cells regardless of the activation condition of the cells ($P < 0.0001$ in either case). The area of Csk clusters in naive cells decreased from 2322 (median, IQR 1310–4522) nm² to 2144 (median, IQR 1261–3770) nm² after stimulation ($P < 0.0001$). This trend was exacerbated for pre-stimulated cells where clusters shrank from 3093 (median, IQR 1632–6154) nm² to 1920 (median, IQR 1030–3655) nm² with stimulation ($P < 0.0001$). Comparing between cell status, the pre-stimulated cells increased their Csk cluster area compared with naive cells for non-stimulatory conditions ($P < 0.0001$) but Csk clusters in pre-stimulated cells became smaller upon stimulation ($P < 0.0001$).

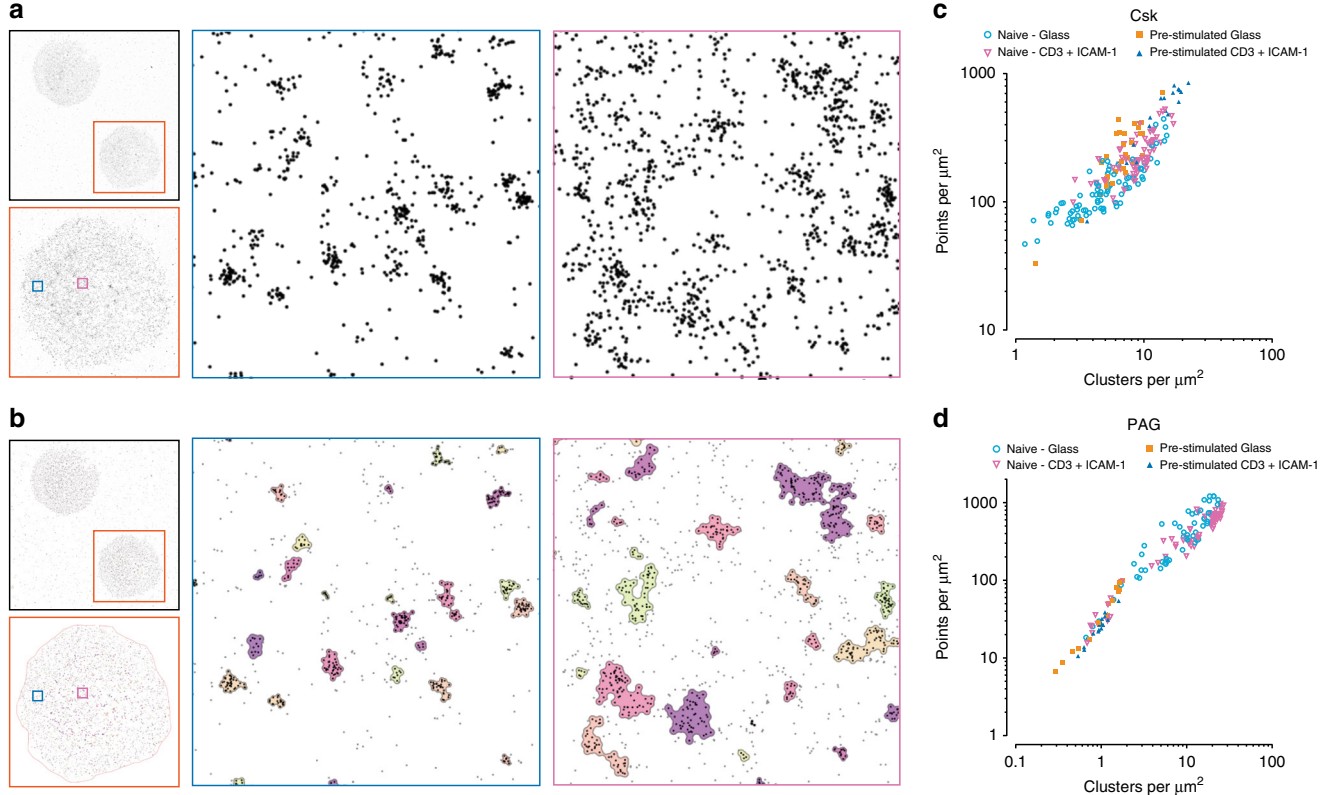

**Fig. 4 Analysis of dSTORM data with Model 87B144.** Activated primary human T cells stained for Csk showing either localized points (**a**), or resulting cluster outlines (**b**), of two regions (blue, peripheral, and magenta, central). Clustering data obtained from individual naive or pre-stimulated cells for Csk (**c**) or PAG (**d**) staining in either non-activating (supported on glass) or activating (anti-CD3 + ICAM-1) conditions. Data are from three independent experiments and are presented as median values from each cell; Csk, naive—glass: 85 cells, Csk, naive—CD3 + ICAM-1: 71 cells, Csk, Pre-stimulated—glass: 30 cells, Csk, pre-stimulated—CD3 + ICAM-1: 19 cells, PAG, naive—glass: 53 cells, PAG, naive—CD3 + ICAM-1: 65 cells, PAG, Pre-stimulated—glass: 14 cells, PAG, pre-stimulated—CD3 + ICAM-1: 18 cells.

PAG clustering (Supplementary Figs. 11g–l and 12e–h) in naive cells did not change between activation conditions for the overall density of clusters, with 10.19 clusters per μm² (median, IQR 5.03–16.07) clusters per μm²) in the non-activated condition and 13.89 clusters per μm² (median, IQR 7.07–21.32 per μm²) for the antibody-activated condition, $P = 0.3303$. However, there was a considerable decrease in the number of PAG clusters, in both non-activated and activated conditions, for pre-stimulated cells compared with naive cells ($P = 0.0002$ for non-activated cells and $P < 0.0001$ for antibody-activated cells). Pre-stimulated cells also showed a decrease in the number of points per cluster between the non-stimulatory condition, with 14 (median IQR 7–32) points per cluster compared with the stimulatory condition with 11 (median, IQR 6–20) points per cluster ($P < 0.0001$). In comparing between the two types of cells there were more points per cluster in pre-stimulated cells than in naive for the non-activating conditions ($P < 0.0001$) but no difference for antigen-activated cells ($P = 0.0763$). Comparing cluster areas between cell types, the pre-stimulated cells had larger PAG clusters (19504 nm², IQR 8597–41690 nm² for non-activated cells and 15669 nm², IQR 8054–30041 nm² for activated cells) compared with naive cells (1551 nm², IQR 841–3159 nm² for non-activated cells and 1359 nm², IQR 766–2522 nm² for activated cells) for both non-stimulatory and stimulatory conditions ($P < 0.0001$ for either case).

**Extension of the model.** As our method utilizes a simple one-dimensional input array of distances, it would be expected to work for 3D data. To test this, we generated simulated data with

similar clustering scenarios as for the 2D data. However, points were also distributed within an additional axial dimension ranging up to 500 nm (again to mimic the axial range encountered in a typical 3D SMLM data set acquired using an astigmatic lens[29]) and clusters were spherical instead of disc shaped. A new model was configured using the same four-layer arrangement as Model XPILJZ (an input layer, two fully connected layers, and an output layer) and set to accept 1000 near neighbor distances. A training data set was prepared as for the other models, except distances were measured in three dimensions. The trained model 'GAXJPR' returned 97.5% accuracy on the testing data set. This model was then used to assess an experimental PALM data set of the T-cell signaling adapter protein LAT fused to mEos3.2 (Supplementary Fig. 13) where it was able to locate clusters of LAT both in the plasma membrane and membrane-proximal vesicles. Although the model was trained on spherical clusters it was still able to identify the elongated clusters that result from the lower axial resolution of astigmatic-based 3D localization imaging.

We also explored the potential of our system to accommodate multiple labels. The models described so far produce a binary label indicating if a point is clustered, with non-clustered points as default. We generated new training data in which points were distributed in either circular clusters, as before, or long filamentous structures. In these data, points therefore had one of three known labels: not clustered, clustered (round), or clustered (fiber). We then trained a model '3TXKFS' (comprising an input layer for data from 1000 neighbors, a fully connected layer, and an output layer to produce a score for each of the three expected labels). Evaluation of novel simulated data confirmed

this model was able to identify circular clusters and fibers with reasonably high accuracy and still performed well when used with experimental data (Supplementary Fig. 14).

## Discussion

In this study, we used neural networks to classify points from SMLM data sets as either clustered or non-clustered, based on a sequence of values derived from each point's nearest-neighbor distances. The network can identify features within such a sequence that allow it to classify points with a high degree of confidence. We have also shown how data sets that have been annotated by our model can then be further processed to use the new information on a point's clustering status to partition points into separate clusters. This process can be performed on the entire available data set, without needing to reduce the data or acquire expensive computational resources. These per-cluster data allow for a deeper interrogation of the clustering patterns within the original image compared with global reporters of clustering, such as Ripley's $K$ Function. This is of relevance to data from biological specimens that, without reduction of the original data to small regions-of interest, is rarely homogenous in its overall distribution of either points or clusters.

With our approach, we observed changes in Csk and PAG clustering in naive and previously stimulated T cells. We saw a marked decrease in the number of PAG clusters for previously activated T cells. The inhibitory function of PAG is not observed in naive T cells, which can be attributed to a more prominent role for Cbl ubiquitin ligase proteins in regulating TCR activation in these cells[30]. Our observations may indicate a change in effectiveness of PAG in capturing Csk at the plasma membrane for Src kinase inhibition after T cells have been exposed to antigen. For example, in naive cells, Csk may have limited PAG-mediated access to the plasma membrane, if PAG were sequestered into clusters or if there was greater competition with other PAG-binding proteins. The overall Csk cluster presentation between naive and pre-stimulated T cells was relatively similar. However, there were detectable changes between non-activated and antibody-activated cells, with Csk clusters having an increased number of points after stimulation accompanied by a very slight decrease in area. This increase in the density of Csk within clusters might reflect the displacement of Csk from its inhibitory, plasma membrane adjacent clusters, to more cytosolic clusters formed with disinhibitory binding partners, such as TRAF3 and PTPN22.

Our approach has several benefits. It is fast, requires minimal parameter input and minimal user interaction compared with other methods. The only analysis parameter that bears any major consequences on the output (and on execution time) is the number of nearest neighbors from which distances are measured. This decision is easily informed from a cursory examination of the input data, for example, by rendering a representative image, identifying a number of candidate clustered structures, and counting the points within them; this can be achieved using the 'restrict to ROI' feature in ThunderSTORM. A model should then be used whose input window (the number of neighbors seen by the model) is greater than the number of points in the target clustered structures and we have also shown that an over-estimation of input size of the model does not have a detrimental effect on the outcome. Underestimation of the input size results in the exclusion of clusters containing more points than the model can 'see'; for samples containing fiducial registration beads (for drift and aberration correction or channel alignment) this could be employed to remove beads, which often present as very large and densely populated clusters within the reconstructed image. Although our models were trained on very simplistic circular clusters, they were also successful at identifying clustered

points a wide variety of non-circular structures. Although there is some latitude for our models to identify exotic clusters, it may be pertinent to specify and train new models on data, which have been simulated to more closely match the type of clusters that are expected in the experimental data. For example, our models are not appropriate to recover clusters containing more points than the models were trained to check. In these cases, for example, a cluster with 200 points will be treated by model 07VEJJ (which only uses 100 nearest-neighbor distances) as a conglomerate of smaller clusters, depending on the internal distribution of points within the larger cluster. Our models were trained on data that included an uneven 'cell' boundary with sharp contrast in the number of points inside and outside of the 'cell' area however points from these areas were not deliberately selected for when building the training set. Data evaluated with our models sometimes shows aberrant clustering within membrane protrusions, but these register as very large and sparsely populated clusters and would be easy to filter out, if desired.

CAML does not make any assumptions about, or corrections for, fluorophore re-blinking. Most SMLM reconstruction software has this function available as part of the image reconstruction post-processing routines and we do not seek to impose any specific blinking correction method upon the user. Other sources of SMLM image artifacts such sample drift, chromatic aberration, imperfect or overlapped point-spread functions, and suboptimal labeling and reporter densities are also assumed to be either absent during image reconstruction or have been accommodated as within acceptable tolerances. Our method also does not consider the points' localization precision as this is a variable that is also dependent on the individual's acquisition and analysis environment; localization precision is dependent on the camera settings (for conversion of intensity values to photon counts) and on the image reconstruction software, which may employ any of a number of different methods to calculate localization precision values. Furthermore, the localization precision is generally much smaller than the size of the structures being detected; in many cases, localization uncertainty values are routinely reported to be below 10 nm. For methods which do incorporate the localization precision, such as Bayesian cluster analysis, removing this parameter (e.g., by giving all points the same value) does not greatly alter the outcome. However, as our system is easily modifiable, it is possible for a user to include a generator of localization uncertainty when simulating new training data and to incorporate this information as an additional feature during model training.

There are few existing methods that apply deep learning to point pattern data sets and none which are specifically designed for SMLM data. PointNet[31] is a supervised neural network architecture, which can classify and segment point-cloud data sets. In PointNet, the network is trained on samples with uniform densities, whereas SMLM data present a very wide range of densities. There are also fundamental differences between data from point clouds and SMLM systems: point clouds describe the external surfaces of objects and points between and within those objects are minimal and often removed by pre-filtering the data. In SMLM data, there are very often many points between (and within) clusters. In some cases, these non-clustered points may comprise most of the data. PointNet has been modified for detection of specific structures, such as caveolae, in SMLM data[32]. However, this required specialized knowledge or high-performance computing facility, whereas CAML can process an entire image without requiring tiling or reduction into small regions-of-interest. A recent theoretical approach to clustering point pattern data is 'dominant sets', which uses pairwise clustering and graph theory to group self-similar points into clusters[33]. This method has been adapted to analyze localization microscopy data of proteins in neuronal synapses[34]; however, it

requires empirical estimation of analysis parameters and computational requirements may limit the number of points that can be assessed.

Although the models described here were trained on simulated clusters with hard-edges and circular shapes, they are nevertheless able to find clusters of abstract and arbitrary shapes, including highly elongated clusters. Our models' assessment of clustering is also not affected by discontinuities in the distribution of points or the type of clustering within the field of view. However, it is important to recognize that bias is inherent in any simulated data set and models trained on such data may not always be appropriate to apply to other types of data. The method as described here is flexible enough to allow an end user to quickly assemble a new model with training data generated to match a particular type of clustering outcome, whereas also allowing rapid re-use of trained models to classify and annotate data for the quantification of clustering in 'real' data.

Future directions include incorporating more features from the input data, such as relative angles to near neighbors or including the localization precision of nearby points, in order to more robustly determine the extent of local point clustering. This will certainly require more sophisticated training data, which are simulated to incorporate these additional features. This information might also be obtained without using simulated data, through manually annotated biological data in which structures of a specific nature were selected. This could be used to train a model to identify those structures in novel data and would also allow the incorporation of features such as localization uncertainty and photon count in their natural context. It may also be possible to use input data other than Euclidean near-neighbor distances. Further improvements could be made if the model were to return output indicating not just if a point originated from a cluster but also which of the nearby points were likely to come from the same cluster. There is also the potential to expand the method into multidimensional data, such as dynamic clustering in live data, or co-clustering between points from different imaging channels.

CAML is deliberately split into discrete processing stages for easier user interaction with each stage but also to allow for intervention between these stages. If a user requires only the model-labeled points, then the processing can be stopped after model assessment. Alternatively, a user may wish to apply a different segmentation method to the labeled data or may want to re-use data prepared by the first step (the most time-consuming stage) with several different models. Each of the data analysis stages of CAML is designed for batch-processing which enables many images to be processed in an unattended fashion, a feature that is absent from some other analysis methods, so it is feasible to automate the output of one stage as input to the next. Another interesting future direction would be to develop the method using unsupervised learning, which would have the advantage of finding clustered structures without being limited or biased towards those given in training data for supervised learning.

The software presented here describes a complementary approach to the existing methods for the cluster analysis of SMLM data. It is presented to be easily accessible to non-experienced users while providing flexibility to enable different and highly customized configurations if required.

## Methods

**Cell isolation and staining**. Peripheral blood was acquired from healthy human donors under ethics license HR-15/16-1978 (King's College London). Written informed consent was provided by each donor. Primary human T cells were isolated from human blood using a pan T-cell selection kit (130-096-535, Miltenyi Biotec). T cells were isolated from human blood and then used directly for imaging assays as "naive" cells. Antigen pre-stimulated T blasts ('pre-stimulated' cells) were generated in parallel by incubating freshly isolated (naive) T cells with 2 μg per ml

anti-CD3 mAb (eBioscience clone OKT3, 16-0037-81) and 5 μg per ml anti-CD28 mAb (RnD Systems, clone CD28.2, 16-0289-85) coated flasks at 50,000 cells per cm² in complete medium (RPMI with 10% fetal bovine serum, L-glutamine, and penicillin/streptomycin) for 2 days. Cells were then washed and cultured in complete medium for an additional 5 days in the presence of 20 ng per mL IL-2 (Proleukin). Purity was then assessed by flow cytometry.

Glass-bottomed chamber slides (#1.5 glass, ibidi μSlides) were coated with a mixture of 3 μg per ml recombinant human ICAM-1-Fc (RnD Systems) and 2 μg per ml anti-CD3 mAb overnight at 4 °C. Cells were added to wells at a density of 25,000 cells per cm² for 4 min then gently rinsed with warm (equilibrated to 37 °C) Hanks' Balanced Salt Solution to remove non-adhered cells then fixed. Fixation was by the pH-shift method[35] at room temperature: cells were first incubated for 5 min in pH 6.8 fixation buffer (3% (w/v) para-formaldehyde (PFA), 80 mM PIPES, 2 mM MgCl$_2$, 5 mM EGTA, in water) followed by 10 min in pH 11 Fixation Buffer (3% (w/v) PFA, 100 mM Borax, in water). Fixed cells were washed three times with PBS, permeabilized with Triton X-100 (0.1% in PBS) for 5 min at 4 °C and rinsed again. Auto-fluorescence was quenched by incubating the sample in NaBH$_4$ (1 mg per ml in water) for 15 min followed by rinsing three times with PBS. The fixed, quenched samples were blocked with 5% (w/v) BSA/PBS for an hour. The sample was then incubated with primary antibody, either rabbit polyclonal anti-Csk (Santa Cruz sc-286) at 1:300 or rabbit polyclonal anti-PAG (Abcam ab14989) at 1:500 overnight at 4 °C and washed three times for 5 min with PBS. The sample was then incubated with secondary anti-rabbit antibody labeled with Alexa Fluor 647 (ThermoFisher Scientific A-21246) at 1:200 for 1 h at room temperature followed by three 5-minute PBS washes. The sample was then used immediately for dSTORM imaging.

For the non-stimulatory condition (referred to as 'glass'), cells were fixed, washed, and stained as a single-cell suspension. Fixed, stained cells were settled onto untreated glass chamber slides prior to imaging.

**dSTORM imaging**. Fixed and stained samples were prepared for imaging by replacing the final PBS wash with a volume of STORM imaging buffer (50 mM Tris-HCl (pH 8.0), 10 mM NaCl, 0.56 M glucose, 0.8 mg per ml glucose oxidase (Sigma G2133), 42.5 μg per ml bovine catalase (Sigma C40), 10 mM cysteamine (Sigma 30070). The dSTORM image sequences were acquired on a Nikon N-STORM system in a TIRF configuration using a 100 × 1.49 NA CFI Apochromat TIRF objective for a pixel size of 160 nm. Samples were illuminated with 647 nm laser light at ~2.05 kW per cm²; no 405 nm laser light was used during imaging. Images were recorded on an Andor IXON Ultra 897 EMCCD using a centered 256 × 256 pixel region at 20 ms per frame for 30,000 frames and an electron multiplier gain of 200 and pre-amplifier gain profile 3.

**dSTORM image reconstruction**. The dSTORM imaging data were processed using ThunderSTORM[36] and the following parameters: pre-detection wavelet filter (B-spline, scale 2, order 3), initial detection by non-maximum suppression (radius 1, threshold at one standard deviation of the F1 wavelet), and sub-pixel localization by integrated Gaussian point-spread function and maximum likelihood estimator with a fitting radius of the pixels. Detected points were then filtered and retained according to the following criteria: an intensity range of 500–5000 photons, a sigma range of 50–250, and a localization uncertainty of < 25 nm. The filtered data set was then corrected for sample drift using cross-correlation of images from five bins at a magnification of 5. The occurrence of repeated localizations, such as can occur from long dye on times or fast re-blinking, was reduced by merging points, which reappeared within 50 nm and 25 frames of the initial detection. For the purposes of interpretation, it is assumed that the frequency of multiple detection of dye molecules is independent of the sample staining and therefore the relative changes in the clustering of points between sample conditions are independent of dye re-blinking.

**Statistical analyses**. For statistical comparisons, all data were analyzed using non-parametric Kruskal–Wallis and Dunn's multiple comparison tests in GraphPad Prism software. A significant difference between conditions was considered as $P < 0.05$ for rejecting the null hypothesis.

**Reporting summary**. Further information on research design is available in the Nature Research Reporting Summary linked to this article.

## Data availability
Simulated data (used for model training and performance evaluation) are available from the Open Science Foundation at https://osf.io/xa4zj/.

## Code availability
The Python scripts and trained models that are described here, as well as a user guide, are available online at https://gitlab.com/quokka79/caml.

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

## Acknowledgements

We acknowledge funding from ERC Starter Grant 337187 to D.O. and BBSRC grant BB/R007365/1 to D.O. S.S. acknowledges support from the Human Frontier Science Program Organization and the Royal Society through HFSP and Dorothy Hodgkin fellowships, respectively. We acknowledge the use of the Nikon Imaging Facility (NIC) at King's College London for data acquisition.

## Author contributions

D.W. conceived the work, wrote and tested the code, acquired and analyzed data, and wrote the manuscript. G.B. prepared and acquired dSTORM data. S.S., R.P., and J.G. provided test data. D.D. conceived the work. D.O. conceived the work and wrote the manuscript.

## Competing interests

The authors declare no competing interests.
