## [Peer Review File · Nature Communications]

Reviewers' comments:

Reviewer #1 (Remarks to the Author):

In this work by Williamson et al., the authors present a machine learning-based clustering method dedicated to point cloud SMLM data, using realistic simulations as a training dataset. While it looks an interesting and trendy approach, this work looks too preliminary to deserve, in the current state, publication in Nature Communication.

There is indeed a major lack of explanations on how things work, what are the important and sensitive parameters, both on the choice of the layers and training datasets, what are advantages, limits and efficiency compared to other popular methods in the field?

The authors choose 3 different learning models, with 2 different architectures and/or input data, without really justifying their choices. It looks like a black box, where the authors use and choose some of the parameters without neither providing more explanations, just assumptions, nor guiding the readers if they aim to reproduce this work on their own datasets.

They use a large number of simulations to train and validate their method, but it is very hard to understand from the chosen range of parameters if the simulation conditions are very challenging and what is the real benefit of using DL approach VS other popular methods? There is an obvious lack of description and representation of the simulation conditions and their relevance with the experimental datasets. The ratio of density between clusters and background should be clearly stated. How close need to be the simulations from the experimental dataset? Is the fluorophore photophysics important to consider? Would the method work the same using PALM, dSTORM or DNA-PAINT without changing any parameters? Are the cell geometry and background noise important to consider in the model? How robust is the method with the model, the simulation parameters and the experimental datasets? Is the pointing accuracy not important to take into account in the simulation?

They claim that one advantage of their method compared to others is that it is parameter free. This is not exactly true since it relies on a very important feature, which is the requirement to simulate a realistic model of the clustering. Moreover, there is an important parameter to the method, the number of neighbors to consider, which looks sensitive to the localization density and should affect both the processing time and memory management.

Are there any specific challenging conditions, such as non-homogeneous background or very heterogeneous datasets, that the proposed method could solve better than standard ones?

The authors mention several times that their approach is faster than existing ones, but never provide any benchmarking that includes the training step.

They only compare their method with other methods on simulation datasets, not on the experimental ones. It is important to be done with some benchmarking. They only use Bayesian inference (very time consuming and model dependent) and G&F (Ripley's function family, slow and parameter sensitive) methods as a comparison. They should also compare with other popular DBSCAN and Tesselation families, which are less sensitive to molecular densities and fast.

The training datasets are all homogeneous clusters with varying sizes and relative cluster-to-background localization densities. But surprisingly, the output of the segmentation performed on experimental data are neither homogeneous in size and shape. This is a bit disturbing, while we could expect the method to split the large clusters in smaller ones to better fit the models. The authors should comment on that, and discuss the relevance of using simulation VS experimental data for training and validation.

Finally, it looks like the method would work directly in 3D, without more complexity and computation time, since it only uses 1D arrays of distances between localized molecules. This would be an interesting advantage over existing methods which usually lack 3D or at the expense of time. This refers to an earlier question about the importance of the pointing accuracy, which in addition would not be isotropic in 3D.

Reviewer #2 (Remarks to the Author):

The paper presents an approach for clustering single molecule localisation (SML) data with a machine learning framework based on a data-driven approaches using models based on Neural Networks (NNs). The model proposed in the paper performs a pre-clustering of the data by solving a classification problem at each point indicating if the point can be clustered or not. Then after this step, a custom-made procedure receives the labelled 2D "point-cloud" and it performs a clustering step identifying which point is associated to each group. Since the reviewer's expertise is mainly related to computational methods, comments will be mainly related to this aspect.

As a general comments learning methods with NNs and Deep NNs (DNN) are very welcome in the field since they can reduce the number of parameters in the clustering procedure and they inherently learn a metric from the data (e.g. embedding) for which the clustering is best performing. On the other end, a data-driven approach should be careful in designing the dataset for training in order to avoid to learn models with restricted generalisation to other experimental domains.

Related to the approach I have three major concerns over the methodology and minor issues related to experimental validation and paper presentation.

- Metric Learning. The strength of NN and DNN approaches is the fact that they can, implicitly or explicitly, learn an embedding for the data which best solves the task at hand. Differently, the proposed approach enforce an Euclidean metric in the first Nearest Neighbors step. This stage might reduce the efficiency of the algorithm since the metric for clustering is imposed by hand instead of having a network learning this step. This happens differently in other models such PointNet [a] that, even if the task is different, accept unordered point clouds (in 3D, but it can be easily applied to the 2D case). To this end, the paper should explain the implications of choosing a specific metric as the Euclidean distance and its impact on the clustering task. Moreover, a discussion about the feasibility of PointNet model for clustering SML should be provided. The best reply to this question would be to come out and testan architecture that do not imposes the Euclidean distance as the metric.

- Data generation and dataset bias. It is true that clustering with NN and DNN reduce the number of parameters. However the model might be biased toward certain data structures given the generated datasets. For this reason, recent methods attempted to avoid to train a clustering method but rather choosing an unsupervised solution [b]. Given the dataset generation procedure presented in the supplemental section, it is clear that the training data favours disk like clusters (line 497) which are pretty common also for previous (non NN and DNN) clustering methods. Moreover the next aggregation step (line 121-129), favour by construction disk like structures. If this approach is to be reusable and more than an application based clustering, it should demonstrate a level of generalisation to other cluster structures. To this end, the authors should show if:

- 1) The post-evaluation cluster segmentation and aggregation step can be eliminated and their model made end-to-end
- 2) If the approach can really generalise to different distributions. The claim in lines 266-268 about generalisation to elongated structures cannot be done without a careful testing on specific structures.

- Experimental evaluation with competing methods. The paper should anyway provides experimental results against DBSCAN as being a very popular clustering method. An evaluation should be also made against the also popular SR-Tesseler [c] and the method presented in [d] using the theory of Dominant Sets for clustering. These papers should be commented in the paper and their parameters optimised to the dataset created in this submission.

Minor comments:

line 64: "Internal configurations of the layers" is ambiguous. The network is fixed, I imagine this being related to parameters optimization.

line 79, paragraph on Model specification: Clarity would be improved by specifying better, as a first step, what is the input and the output of the network. The description is there but it can be made more effective and understandable.

line 92: "N" is a parameter of the overall approach. Even if later there is a sentence indicating that the algorithm is robust to choices of N, but a more complete experimental evaluation of this aspect should be provided in the supplemental material (e.g. when the methods broke down for wrong choices of N).

line 117: make uniform the word "dataset", sometimes is named data-set or data sets, etc. Choose one name and stick with it.

line 122-129: There should be an indication why the procedure chosen for creating clusters is best.

Supplementary Figure 1: the fonts in the plots are barely visible

[a] Qi, Charles R., et al. "Pointnet: Deep learning on point sets for 3d classification and segmentation." Proceedings of the IEEE Conference on Computer Vision and Pattern Recognition. 2017.

[b] Ji, Pan, et al. "Deep subspace clustering networks." Advances in Neural Information Processing Systems. NeurIPS 2017.

[c] Levet, Florian, et al. "SR-Tesseler: a method to segment and quantify localization-based super-resolution microscopy data." Nature methods 12.11 (2015).

[d] Pennacchietti, Francesca, et al. "Nanoscale molecular reorganization of the inhibitory postsynaptic density is a determinant of GABAergic synaptic potentiation." Journal of Neuroscience 37.7 (2017).

Reviewers' comments:

Reviewer 1

Reviewer #1 (Remarks to the Author):

In this work by Williamson et al., the authors present a machine learning-based clustering method dedicated to point cloud SMLM data, using realistic simulations as a training dataset. While it looks an interesting and trendy approach, this work looks too preliminary to deserve, in the current state, publication in Nature Communication.

There is indeed a major lack of explanations on how things work, what are the important and sensitive parameters, both on the choice of the layers and training datasets, what are advantages, limits and efficiency compared to other popular methods in the field? The authors choose 3 different learning models, with 2 different architectures and/or input data, without really justifying their choices.

The aim of this work is to demonstrate that machine learning can be applied to the problem of classifying points in a typical SMLM dataset. We have shown several learning models with different configurations and performances to demonstrate this flexibility; the idea being that users can take our code and trained models (all of which are available on the stated GitLab repository) and use them with their data, or they can create entirely new layouts without needing to change large parts of the underlying code.

The variety of model architectures that can be created in Keras is enormous and therefore we chose to describe two architectures which worked well: a basic network and a more complicated setup which showed greater classification accuracy. The framework we have created allows for flexibility in the model configuration (as in the assembly of layers) and the input data. We now make this clearer in the manuscript and provide some guidance as to why we chose to demonstrate the method with the architectures that we did, and how the user might design new architectures to suit their own data. We have also added text as the reviewer suggests, discussing what are the important or sensitive parameters that the user should be aware of.

There are no other machine learning based cluster analysis methods for SMLM data currently available. Compared to non-ML methods, we have performed additional analysis to show comparisons versus the popular methods of DBSCAN and Voronoi tessellation. Our method shows clear advantages in classification accuracy, flexibility, lack of user-induced bias though strongly influential analysis parameters and computation advantages in terms of processing speed.

It looks like a black box, where the authors use and choose some of the parameters without neither providing more explanations, just assumptions, nor guiding the readers if they aim to reproduce this work on their own datasets.

The code is designed such that the learning models we demonstrate here can be used with any other clustered SMLM data. We demonstrate the accuracy of the models and parameters on a wide variety of data sets mimicking realistic SMLM data. All code and training data are available on the Github site, so readers can easily reproduce the results on their data. Taking the reviewers advice, we now provide more guidance on the effect choice of parameters and training data, should users wish to train the models on specific training data sets related to their own work e.g. if their experimental data sets are unusual compared to what is available in the literature. We extend this discussion to the choice of models and architecture should readers want to go further and build new architectures tailored for their data.

They use a large number of simulations to train and validate their method, but it is very hard to understand from the chosen range of parameters if the simulation conditions are very challenging and what is the real benefit of using DL approach VS other popular methods?

The simulation conditions are chosen to cover a broad range of clustering conditions typically found in the SMLM literature. Where conditions are naturally bounded (e.g. percent of points in the image being in clusters bounded between 0% and 100%) we cover the whole range. For unbounded conditions, e.g. number of points per cluster, we have restricted these parameters to those which might be expected in a typical biologically derived image, and these are now stated clearly in the Supplementary Information. Combining these, we therefore demonstrate the methodology on a huge range of conditions, centred on those which are realistic but extending out to even unrealistically challenging conditions (e.g. with very little clustering). Here again, we now include new analysis to compare our approach with other commonly used cluster analysis methods for SMLM – DBSCAN and Voronoi tessellation. Both these are heavily influenced by strong analysis parameters which cannot be well selected by the user and we now discuss the advantages of our method in terms of classification accuracy, flexibility, robustness and computational efficiency.

There is an obvious lack of description and representation of the simulation conditions and their relevance with the experimental datasets.

We now include a detailed description of all the simulated conditions use, detailing every parameter of these conditions such that they can be perfectly reproduced by the reader. This information is in the Supplementary Information. Additionally, we have now included a section of discussion on which subsets of these represent “realistic” conditions likely to be encountered in experimental SMLM data sets and which represent more challenging conditions designed to probe the ultimate limitations of the machine learning methodology.

The ratio of density between clusters and background should be clearly stated.

As it was, this ratio would need to be calculated by the reader from the stated simulation parameters. However, we have now taken the Reviewer’s advice and provided this ratio for all conditions in the Supplementary Information (Supplementary Table 2).

How close need to be the simulations from the experimental dataset?

We have used training data generated by simulations which were designed to roughly mimic experimental SMLM data sets e.g. in terms of cluster shapes and profiles, density of localisations and so on. Models trained using these simulations are then used to analyse a wide range of simulated data (as mentioned above). Those conditions which are realistic for SMLM experiments show high accuracy, however, we also included less realistic simulations to test the limits of these trained models. These can therefore be used to test the accuracy of classification for a given simulated data set when the model was taught on a different simulated condition.

Is the fluorophore photophysics important to consider?

This method makes no account for the resampling of points due to label reactivation (re-blinking) as the assumption exists that the input data have already been corrected for this in the post-processing stages of image reconstruction, e.g. by identifying re-blinking points and combining them into one, after Annibale et al. (2011). This is a feature available in almost all SMLM reconstruction software including ThunderSTORM, which we use here. We now point this out more clearly in the text. Should a user be interested in the effect of the re-blinking correction method they can always compare uncorrected vs corrected data with our software. There are other experimental effects

which are also assumed to be correct by our code e.g. that not all proteins will be labelled, sample drift during the acquisition time, non-perfect point-spread-functions due to imperfect optics and errors of localisation due to overlapped PSFs in a single frame. Again, many of these are automatically corrected for in most common reconstruction software (like ThunderSTORM) but we now discuss these explicitly in the text.

Would the method work the same using PALM, dSTORM or DNA-PAINT without changing any parameters?

Provided that the clusters within the data have fewer points than the number of nearest neighbours sampled for the model, then the method should find those clusters. The usual rule applies that for any SMLM experiment it is important to consider the goal for analysis and select an imaging approach accordingly. For example, PALM is better suited to quantification as the labelling is specific, efficient, and there is minimal re-blinking. Any re-blinking events are readily identifiable for merging, post-processing. Imaging with dSTORM and DNA-PAINT are best suited when the goal is to determine structural components with the sample, wherein re-blinking can be used to effectively over-sample underlying structures. Oversampling can also be exploited in order to discard all but the best points (in terms of brightness and localization uncertainty). Ultimately however, yes, the method is applicable to PALM, STORM and DNA-PAINT, and we now state this explicitly.

Are the cell geometry and background noise important to consider in the model?

Cell geometry can indeed have an important impact on any cluster assessment method; for TIRF derived images, there is an assumption that the plasma membrane is generally relatively flat against the glass although sharp deformations in the membrane could give an impression of clustering. Such deformations can occur with membrane ruffling or ciliated structures which are trapped between the cell and the glass, but the scale of these features is usually much greater than that of the protein clusters which we are interested in. At the boundary of the cell there can be artefacts as well, where the plasma membrane curves around and away from the glass surface. All these artefacts are consequences of the imaging methods, where a very thin but nonetheless three-dimensional volume of emitters is projected onto a two-dimensional sensor.

Very thin regions of the cell, such as filopodia in contact with the glass, can register as highly clustered areas. These areas may contain the same point pattern as the rest of the cell but as they are judged relative to the sparsely populated space outside of the cell, as such non-clustered points here can be incorrectly classified as clustered. We now discuss both points in the manuscript text.

Background noise, as in signal which is not specific to the actual target, should ideally be minimized in any SMLM experiment. When such signal is present, e.g. due to non-specific binding of antibody to the glass, or antibody aggregates, then these will be detected and classified like any other point from the image. In any case, our method is intended to be used on the entire image first, with clustering information retrieved from regions which are drawn after classification and cluster segmentation. In this way, it is possible for a user to compare the clustering in the 'cell' regions to that in the 'non cell' regions and therefore identify the characteristics of clusters which may come from non-specific (background) interactions – if the user had particularly noisy or low quality data.

How robust is the method with the model, the simulation parameters and the experimental datasets?

We have demonstrated the robustness of the methods by analysing hundreds of simulated data sets with character ranging from realistic, to much more challenging as well as by analysing simulated data. We have now, additionally, tested the method in comparison to other popular cluster analysis

methods used for SMLM data. We show that for realistic data, our method delivers more accurate results than other techniques and we fully map the limits of applicability. As stated, we now also give guidance on how the model and simulation parameters were selected both as justification, and to allow users to modify the methodology should they have unusual data sets, unlike those found in the literature and therefore covered by our range of simulations.

Is the pointing accuracy not important to take into account in the simulation?

The precision with which points are localized is not currently used in the analysis. The reason for this is as follows: In general, the localisation precision is much smaller than the size of objects being detected. For example, the most recent papers report localisation precisions down below 10 nm. Secondly, methods which do use the localisation precision, such as Bayesian cluster analysis (2015), show that including it has a minimal effect on the accuracy of analysis and 3) localisation precisions are calculated for each molecule individually and are a theoretical estimate e.g. based on camera design, number of collected photons etc. There are a number of different methods of calculating the precision in the field and the best method is not agreed upon. That said, it is possible to construct the machine learning approach to use the localisation precision as an additional vector and we have added text to this effect in the discussion.

They claim that one advantage of their method compared to others is that it is parameter free. This is not exactly true since it relies on a very important feature, which is the requirement to simulate a realistic model of the clustering. Moreover, there is an important parameter to the method, the number of neighbors to consider, which looks sensitive to the localization density and should affect both the processing time and memory management.

We do not claim the method is parameter free and this phrase is not in found within the text; we say the method “requires minimal parameter input”. The number of near neighbours is an important consideration and that it does indeed affect processing time and memory management – this is stated in the text. Importantly however, this parameter is a balance between computational time and potential artefacts, and therefore can always be set conservatively – in cases where it does not impact the analysis. We have also found that the models described here are not especially sensitive to the type of clusters presented during training and that as long as the model sees enough points from a wide enough variety of clustering patterns then it is able to classify points within clusters from real data, including clusters that were not available to the model during training. We conclude therefore that the method is robust and can correctly identify cluster morphologies that it has not encountered in training. We have now included discussion of this point in the text.

Are there any specific challenging conditions, such as non-homogeneous background or very heterogeneous datasets, that the proposed method could solve better than standard ones?

Our method seems capable of handling inhomogeneous clustering and performs acceptably on strongly varying backgrounds of non-clustered points. We have included new data which contains a variety of different clustered structures on an undulating background to demonstrate this.

The authors mention several times that their approach is faster than existing ones, but never provide any benchmarking that includes the training step.

We have now included a full quantitative analysis of the computational time required by our method in comparison to Bayesian cluster analysis and DBSCAN, popular alternatives in the field. We show that our method is faster than these approaches for very large sets of data.

They only compare their method with other methods on simulation datasets, not on the experimental ones. It is important to be done with some benchmarking. They only use Bayesian inference (very time consuming and model dependent) and G&F (Ripley's function family, slow and parameter sensitive) methods as a comparison. They should also compare with other popular DBSCAN and Tessellation families, which are less sensitive to molecular densities and fast.

We agree with the reviewer that DBSCAN and Tessellation (such as SR-Tesseler) are popular alternative methods used for SMLM data. We have therefore now included a comparison of our method with these techniques using experimental data, as the reviewer suggests.

The training datasets are all homogeneous clusters with varying sizes and relative cluster-to-background localization densities. But surprisingly, the output of the segmentation performed on experimental data are neither homogeneous in size and shape. This is a bit disturbing, while we could expect the method to split the large clusters in smaller ones to better fit the models. The authors should comment on that, and discuss the relevance of using simulation VS experimental data for training and validation.

We do not expect to recover perfectly circular cluster boundaries. Clustered data are simulated such that points within a cluster are distributed within a circular boundary linked to the cluster radius parameter. This boundary represents the maximum distance at which points can be distributed from the centre of the cluster. Therefore, and especially for larger but sparsely populated clusters, the points within a cluster are unlikely to ever describe a neat disc around the cluster centre. The subsequent construction of cluster outlines does not attempt to force any type of shape upon the clusters and the ultimate cluster outline is derived from the spatial distribution of the clustered points which comprise the cluster. For clusters which were described as very large and very dense then certainly we would expect see a cluster outline returned by the software which was more-or-less circular. In conclusion then, the simulated cluster radius is the maximum possible extent of the cluster, whereas the analysis returns the actual extent of the points that make it up. We have now modified the text and included a new figure (Supplementary Figure 1) to make this process clear.

Finally, it looks like the method would work directly in 3D, without more complexity and computation time, since it only uses 1D arrays of distances between localized molecules. This would be an interesting advantage over existing methods which usually lack 3D or at the expense of time. This refers to an earlier question about the importance of the pointing accuracy, which in addition would not be isotropic in 3D.

We thank the reviewer for this suggestion and agree that the method is applicable to 3D data. We have now performed machine learning cluster analysis on 3D data and included this demonstration in the manuscript.

Reviewer 2

Reviewer #2 (Remarks to the Author):

The paper presents an approach for clustering single molecule localisation (SML) data with a machine learning framework based on a data-driven approaches using models based on Neural Networks (NNs). The model proposed in the paper performs a pre-clustering of the data by solving a classification problem at each point indicating if the point can be clustered or not. Then after this step, a custom-made procedure receives the labelled 2D "point-cloud" and it performs a clustering step identifying which point is associated to each group. Since the reviewer's expertise is mainly related to computational methods, comments will be mainly related to this aspect.

As a general comments learning methods with NNs and Deep NNs (DNN) are very welcome in the field since they can reduce the number of parameters in the clustering procedure and they inherently learn a metric from the data (e.g. embedding) for which the clustering is best performing. On the other end, a data-driven approach should be careful in designing the dataset for training in order to avoid to learn models with restricted generalisation to other experimental domains. Related to the approach I have three major concerns over the methodology and minor issues related to experimental validation and paper presentation.

- Metric Learning. The strength of NN and DNN approaches is the fact that they can, implicitly or explicitly, learn an embedding for the data which best solves the task at hand. Differently, the proposed approach enforce an Euclidean metric in the first Nearest Neighbors step. This stage might reduce the efficiency of the algorithm since the metric for clustering is imposed by hand instead of having a network learning this step. This happens differently in other models such PointNet [a] that, even if the task is different, accept unordered point clouds (in 3D, but it can be easily applied to the 2D case). To this end, the paper should explain the implications of choosing a specific metric as the Euclidean distance and its impact on the clustering task.

We decided to use near-neighbour Euclidean distances as the feature vector for this paper as it allows us to interrogate each point in turn (thus removing any dependency on order from the original input data) and because a fixed number of neighbours allows for a consistently shaped input vector. Performing the initial extraction and normalisation of near-neighbour distances from the raw data is therefore an elementary first step in feature engineering. Moreover, it is intuitive to the class of questions posed in the biological sciences as Euclidean distances can be compared to other biological measures of potential interest such as cell size, cell speeds, tissue architecture and so on, which are not described by point clouds.

PointNet trains on a fixed number of points (2048 points for training and the same number of points, randomly sampled from ModelNet40, for testing) from each training mesh when learning the features to classify objects in a point cloud. This is a very different situation to our data where we do not expect our clusters to be described by so many points and where the number of points can be very large and highly variable between images; subsampling 4096 points from a set of 1 million would readily result in missed clusters, hence our desire to process all the available data. Furthermore, the point-cloud data describes the objects surfaces (the external, light-reflective boundary) whereas our clusters contain points located within the cluster itself. Also, there is very little (to no) noise between and within objects because the points in a point cloud dataset almost always belong to an object surface, whereas SMLM data can often feature a large non-clustered population arising from both specific signal (monomers of the target protein) and non-specific signals (e.g. off-target antibody staining).

Moreover, a discussion about the feasibility of PointNet model for clustering SML should be provided. The best reply to this question would be to come out and test an architecture that do not imposes the Euclidean distance as the metric.

Thank-you for this suggestion; we have investigated the possibility of testing an architecture which does not impose Euclidean distances. Two new models, each based around the two configurations presented in the paper, have been created using only the normalized spatial coordinates of 1000 nearest neighbours as input. A model built around the simpler 4-layer configuration achieved comparable performance (approximately 94% classification accuracy) to the equivalent trained on distance-derived values. For the more complicated 11-layer arrangement (employing LSTMs) the model also achieved equivalent performance (about 93% classification accuracy) and was considerably slower than its distance-fed sibling taking over two weeks to train. While it may be possible to further refine these models to improve their performance, we feel that there is not a great

deal to be gained by not using the Euclidean distance metric, especially as this allows trivial extension of the method into three spatial dimensions without changing the underlying input structure. Nevertheless, as a significant amount of processing time is spent identifying each point's nearest neighbouring point, it might be possible to use methods which return the approximate nearest neighbour points (e.g. ANNOY) as input, to improve the overall performance. A discussion on these matters and the non-Euclidean trained models is now included in the main text.

PointNet has been adapted to analyse specific structures from super-resolution data (Khater et al, 2018) however, it required pre-processing steps including data filtering and pre-segmentation. Computational load was also an issue, with the method run in an HPC facility. For this reason, we don't believe the development of an architecture and its direct comparison to PointNet is feasible. Instead, we have included an extended discussion of PointNet in our manuscript, making the user aware of it as an alternative method and the potential advantages and disadvantages of each technique.

- Data generation and dataset bias. It is true that clustering with NN and DNN reduce the number of parameters. However the model might be biased toward certain data structures given the generated datasets. For this reason, recent methods attempted to avoid to train a clustering method but rather choosing an unsupervised solution [b].

We have investigated the method in [b] but found it to not really be appropriate for analysing biological data derived from SMLM. However, we agree with the reviewer that attempting to develop unsupervised methods for SMLM data analysis might be possible. These methods are not as well developed as the supervised versions (hence our choice) and therefore are currently somewhat untested for these applications. However, we have now included additional discussion in the text that such methods could be attempted and developed in future, to attempt to avoid potential bias.

Given the dataset generation procedure presented in the supplemental section, it is clear that the training data favours disk like clusters (line 497) which are pretty common also for previous (non NN and DNN) clustering methods. Moreover the next aggregation step (line 121-129), favour by construction disk like structures. If this approach is to be reusable and more than an application based clustering, it should demonstrate a level of generalisation to other cluster structures. To this end, the authors should show if:

1) The post-evaluation cluster segmentation and aggregation step can be eliminated and their model made end-to-end

Our motivation for this strategy is that the questions posed in the biological sciences are quite diverse. Biological researchers might want to know about the un-clustered (monomeric) points, cluster sizes or complex shape descriptors, the spatial relationship between clusters and so on. We therefore build a workflow which is largely modular with the cluster segmentation step working on the output of the machine learning stage. This could then be modified to answer other biological questions without a complete redesign of the workflow. That said, we agree it is possible to build a ML model to work end-to-end and we now discuss this possibility in the manuscript.

2) If the approach can really generalise to different distributions. The claim in lines 266-268 about generalisation to elongated structures cannot be done without a careful testing on specific structures.

We have now included new analysis on the generalisation to other cluster morphologies.

- Experimental evaluation with competing methods. The paper should anyway provides experimental results against DBSCAN as being a very popular clustering method.

As the reviewer suggests (and similarly to Reviewer 1) we have now included a comparison to other methods including DBSCAN. We show that our method has advantages of accuracy, flexibility, robustness to user-defined analysis parameters and computational speed.

An evaluation should be also made against the also popular SR-Tesseler [c] and the method presented in [d] using the theory of Dominant Sets for clustering. These papers should be commented in the paper and their parameters optimised to the dataset created in this submission.

We agree with the reviewer that, like DBSCAN, we should have included comparisons to SR-Tesseler which is indeed popular in the field. Like DBSCAN, such comparisons are now included.

Minor comments:

line 64: "Internal configurations of the layers" is ambiguous. The network is fixed, I imagine this being related to parameters optimization.

We agree with the reviewer and have made this clear.

line 79, paragraph on Model specification: Clarity would be improved by specifying better, as a first step, what is the input and the output of the network. The description is there but it can be made more effective and understandable.

We have added a discussion on the input and output of the network, as suggested.

line 92: "N" is a parameter of the overall approach. Even if later there is a sentence indicating that the algorithm is robust to choices of N, but a more complete experimental evaluation of this aspect should be provided in the supplemental material (e.g. when the methods broke down for wrong choices of N).

We have included an evaluation of the incorrect choice of N, i.e. where the number of points in a cluster exceeds the input size of the model. Specifically, we show the results for model 07VEJJ (N=100 input) where clusters have 150 or 200 points per cluster (see Supplementary Figure 4).

line 117: make uniform the word "dataset", sometimes is named data-set or data sets, etc. Choose one name and stick with it.

Thank-you, this has been corrected.

line 122-129: There should be an indication why the procedure chosen for creating clusters is best.

We have now included this discussion. The method was chosen for simplicity, because it can generate well defined cluster descriptors for comparison to the output and because of its ability to recapitulate realistic SMLM data sets.

Supplementary Figure 1: the fonts in the plots are barely visible

We have increased the font-size throughout this (and other) figures to improve legibility of the text.

Reviewers' comments:

Reviewer #1 (Remarks to the Author):

In this new revision, Williamson et al. answered most of my concerns. I'm quite satisfied with this new version of the manuscript and support its publication in Nature Communication after answering the following minor points:

- Computation time should be more precisely detailed, maybe in a specific section in the main text. It is indeed also very important to inform the readers about the time required for training for each scenario and network. In the actual version of the manuscript, there is a vague mention in the supplemental about hours to days for training on some possible scenarios. This needs to be clearly visible and written in the benchmarking section. Even if it has to be done only once, we know that in practice, training is empirical and usually requires several iterations before succeed. Maybe there would be a way to use the pretrained network and refine the coefficients on new simulation data to speed-up the training process?
- Comparison with alternative methods, eg. G&F, Bayesian clustering, DB-SCAN and SR-Tesseler, should be better synthetized, both for the cluster analysis efficiency and the computation time. In the current version, all scenarios are not compared with all the methods, making it confusing. Same remark for the computation time, where all the methods with no exception should appear on the graph (stating for example that SR-Tesseler requires between 1-5 minutes for all the simulations is not really acceptable; even if it requires some user input, which is the case for all the methods).
- After verification in the Gitlab, it looks like the simulations used for the paper are not available. Even if the authors provide the script to generate simulations, which is very important, it is also important to provide the simulations used for the training and validation.
- In their rebuttal, the authors state that the localization precision has a minimal impact on the quantifications (similar to Bayesian cluster analysis) and is therefore not considered in CAML. This should be clearly stated and explained in the text.
- The results in Supp Fig 12 look really promising, but I don't really understand why it works so well? This is indeed somewhere in contradiction with results in Supp Fig 2, which shows that it is possible to segment elongated structure even when trained with circular clusters. Is there an explanation, eg. the density between the fibers and the clusters different? The authors should discuss these results in the manuscript.

Reviewer #2 (Remarks to the Author):

The revised manuscript provides further insight related to the experimental validation related to the competing, unsupervised approaches, namely DBSCAN and SR-Tesseler. Both these methods are surpassed by [d] in most scenarios and a further attempt to compare with such approach (e.g. not implementing the approach from scratch but possibly requesting code).

The authors put indeed a lot of efforts in replying the (several) reviewers comments and for most of the points they succeeded in providing convincing answers. Yet I see still two points that requires further explanation and experimental validation.

Both reviewers had a similar concern expressed in the question:

R1: " [...] Moreover, there is an important parameter to the method, the number of neighbors to consider, which looks sensitive to the localization density and should affect both the processing time and memory management."

R2: "N" is a parameter of the overall approach. Even if later there is a sentence indicating that the algorithm is robust to choices of N, but a more complete experimental evaluation of this aspect should be provided in the supplemental material (e.g. when the methods broke down for wrong choices of N).

Where authors did reply with the following sentences:

"The only analysis parameter which bears any major consequences on the output (and on execution time) is the number of nearest neighbors from which distances are measured. This decision is easily informed from a cursory examination of the input data and an overestimation of this parameter does not have a detrimental effect on the outcome."

And

"We have included an evaluation of the incorrect choice of N, i.e. where the number of points in a cluster exceeds the input size of the model. Specifically, we show the results for model 07VEJJ (N=100 input) where clusters have 150 or 200 points per cluster (see Supplementary Figure 4)."

While the first authors reply confirm that N is a parameter that "bears major consequences", they then indicate that the choice of N can be done by, a not very well defined, "cursory examination of the input data". For this issue, the reviewer need a higher amount of details on how this examination can be done and how much this procedure is subjective and error-prone. The second author reply is instead unsatisfactory, it is hard to judge, from a single experiment (i.e. N=100) how the choice of the number of nearest neighbours affect performance. Since this parameter is so important for the approach a deeper analysis is necessary to understand its influence. On synthetic data this can be easily done since ground truth is available. At increasing values of N it is possible to evaluate Precision, Recall and F1. This test is necessary in order to reply to our previous questions. Moreover, another qualitative evaluation on real data should be presented at different values of N to grasp the real differences in changing the parameter.

The next major aspect that requires clarification and balancing in the paper is still related to the dataset bias. It is a fact in the machine learning community that any synthetic dataset brings implicitly a bias, which is related to the different statistics between real and generated data. Sentences in the paper like:

"Although the models described here were trained on simulated clusters with hard edges and circular shapes, they are nevertheless able to find clusters of abstract and arbitrary shapes, including highly elongated clusters"

are strong statement even if supported by some experimental evidence (see later). Even more because it is clear that the method is synthetic data dependent as hinted by the very same authors when saying:

"For example, our models are not appropriate to recover clusters containing more points than the models were trained to check"

Here there are then two requests. Balance the paper in its conclusions, there will always be a gap between real and synthetic data unless different methods are used (e.g. GAN). Be also precise over the capability of the network of generalising on elongated and other clusters. The paper mention a "A qualitative assessment" when evaluating elongated structures but the question is why was not possible to do a quantitative evaluation to measure the recall, precision and F1 over

the recovered elongated clusters. It is very useful to understand which is the drop of performance in using clusters of different shape (it has to be a drop, validation on them possibly is not performed) and analyse in depth this data.

As a general comment, the conclusion of the paper should state clearly the shortcomings to have a balance judgement of this work related to:

- Synthetic dataset bias
- Drop in performance when considering different cluster arrangements
- The parameter N has an influence on results
- Euclidean norm is a design choice (with its own properties) but it might not be the best metric with other cluster rearrangements (especially in 3D)

Reviewer #1 (Remarks to the Author):

In this new revision, Williamson et al. answered most of my concerns. I'm quite satisfied with this new version of the manuscript and support its publication in Nature Communication after answering the following minor points:

- Computation time should be more precisely detailed, maybe in a specific section in the main text. It is indeed also very important to inform the readers about the time required for training for each scenario and network. In the actual version of the manuscript, there is a vague mention in the supplemental about hours to days for training on some possible scenarios. This needs to be clearly visible and written in the benchmarking section. Even if it has to be done only once, we know that in practice, training is empirical and usually requires several iterations before succeed. Maybe there would be a way to use the pretrained network and refine the coefficients on new simulation data to speed-up the training process?

Computation times for CAML have been added to the SI for every stage of processing; Supplementary Table 3 contains indicative training times for data simulation, preparation, and model training and Supplementary Figure 8 contains times for novel data evaluation. We have also stated how these times can be improved, e.g. using early stopping to halt training once validation accuracy is no longer increasing. The bulk of the computation time involves constructing distance matrices for all points in the training images dataset ('Stage 1') and a lot of the time is down to I/O transfer. This approach was chosen as it offers flexibility during development. However, time can be saved if candidate training examples were selected during the data simulation stage ('Stage 0'), and distance matrices created on just those points (instead of on all points) for pooling into the training, validation, and testing datasets. This alternative approach will be included as an option in the code for a future release.

It is entirely possible within the Keras framework to take a trained model and perform additional training, e.g. should more training data become available. However, we anticipate that most of the work done when training models will involve changes which would require a new model to be trained, for example changes to the model's configuration, e.g. changing the specification or arrangement of layers or adding or removing layers, or changes to the format of the input data, e.g. the number of neighbours or classification labels.

- Comparison with alternative methods, eg. G&F, Bayesian clustering, DB-SCAN and SR-Tesseler, should be better synthesized, both for the cluster analysis efficiency and the computation time. In the current version, all scenarios are not compared with all the methods, making it confusing. Same remark for the computation time, where all the methods with no exception should appear on the graph (stating for example that SR-Tesseler requires between 1-5 minutes for all the simulations is not really acceptable; even if it requires some user input, which is the case for all the methods).

We don't believe it is feasible to compare every simulated condition with every other cluster analysis method due to space and publishing requirements. However, we have compared all available alternative methods on a 'baseline' scenario, presented in the accuracy heatmaps in Fig 3a (for CAML), Fig 3b (for G&F), Fig 3c (for Bayesian) and Supplementary Figure 5 (DBSCAN). Where we have shown other clustering scenarios (as in Fig. 3d, e, and f) then this is to compare the performance of that method across different scenarios rather than to compare the scenario across different methods.

Regarding SR-Tesseler, the only information which can be output from SR-Tesseler are the number of clusters that it found and each cluster's shape descriptors: area, point-count, and some other geometry measurements. In order to create the accuracy heatmaps we would need to recover the list of input points, their original labels, and the labels assigned by SR-Tesseler. As this information is not

available in the published software it is impossible to create accuracy heatmaps for SR-Tesseler. We therefore worked with the information that we do have access to and plotted cluster area *vs* points-per-cluster (Supplementary Fig. 6) between CAML and SR-Tesseler. Furthermore, as SR-Tesseler is not able to batch-process data it is impractical to compare all 1,219 images in the baseline scenario manually, so we selected nine images representing the extreme and central elements of the baseline scenario for comparison to CAML.

We have added timings for SR-Tesseler to Supplementary Figure 8. We originally decided not to include these because they are highly subjective; for example, although SR-Tesseler is computationally faster, it still requires the user to be present throughout the process. CAML, DBSCAN, and Bayesian only require user intervention at the beginning but can then run unattended.

- After verification in the Gitlab, it looks like the simulations used for the paper are not available. Even if the authors provide the script to generate simulations, which is very important, it is also important to provide the simulations used for the training and validation.

We appreciate the Reviewers support of open science and data availability. Due to space restrictions on GitLab it is not possible to keep all the original input data in the same repository as the code. We have now made the original training and evaluation datasets available through the Open Science Foundation at <https://osf.io/xa4zj/> and updated the manuscript and the GitLab wiki with this information. We have also updated the code to instruct users on how to set the seed for Python's pseudorandom number generator; this should allow users to generate completely identical number sequences and therefore be able to reconstruct the exact original training image files.

- In their rebuttal, the authors state that the localization precision has a minimal impact on the quantifications (similar to Bayesian cluster analysis) and is therefore not considered in CAML. This should be clearly stated and explained in the text.

As requested, we have added text to the manuscript making this point clear.

- The results in Supp Fig 12 look really promising, but I don't really understand why it works so well? This is indeed somewhere in contradiction with results in Supp Fig 2, which shows that it is possible to segment elongated structure even when trained with circular clusters. Is there an explanation, eg. the density between the fibers and the clusters different? The authors should discuss these results in the manuscript.

The data presented in these two figures are for two different models. In Supplementary Figure 2 we show an artificial dataset containing a mixture of different sizes and types of clusters against an uneven background of non-clustered points. This image was created from an image mask so there is no underlying ground-truth information. This image was assessed with Model 87B144, which was only ever trained on circular disc-shaped clusters. The resulting images show which points were classified as clustered or not clustered, but such a model cannot further separate clustered points into round-clustered or fibre-clustered.

This contrasts with Supplementary Figure 14 (formerly SI Fig 12), where new training data were created which contained points that were either 'fibrous-clustered', 'round-clustered', or 'not-clustered'. Such data were then used to train a new model which *could* specifically identify points as belonging to one of the three classes. The evaluated data scatterplots can therefore also be coloured to indicate these different labels.

Reviewer #2 (Remarks to the Author):

The revised manuscript provides further insight related to the experimental validation related to the competing, unsupervised approaches, namely DBSCAN and SR-Tesseler. Both these methods are surpassed by [d] in most scenarios and a further attempt to compare with such approach (e.g. not implementing the approach from scratch but possibly requesting code).

As stated for Reviewer 1, we have compared our method to all comparable publicly available alternative SMLM cluster analysis methods. Reference [d] that the Reviewer refers to do not make their code available, nor say in what language it was implemented or what computational resources/times are required. Further, from the description, the method requires a construction of a full distance matrix, limiting the analysis to fewer than tens of thousands of points (e.g. to avoid exceeding available memory in a typical workstation) and the method requires analysis parameters which must be empirically estimated. For these reasons this comparison is simply not possible. We do however discuss and reference this method in our manuscript.

The authors put indeed a lot of efforts in replying the (several) reviewers comments and for most of the points they succeeded in providing convincing answers. Yet I see still two points that requires further explanation and experimental validation. Both reviewers had a similar concern expressed in the question:

R1: “ [...] Moreover, there is an important parameter to the method, the number of neighbors to consider, which looks sensitive to the localization density and should affect both the processing time and memory management.”

R2: "N" is a parameter of the overall approach. Even if later there is a sentence indicating that the algorithm is robust to choices of N, but a more complete experimental evaluation of this aspect should be provided in the supplemental material (e.g. when the methods broke down for wrong choices of N).

Where authors did reply with the following sentences:

“The only analysis parameter which bears any major consequences on the output (and on execution time) is the number of nearest neighbors from which distances are measured. This decision is easily informed from a cursory examination of the input data and an overestimation of this parameter does not have a detrimental effect on the outcome.”

“We have included an evaluation of the incorrect choice of N, i.e. where the number of points in a cluster exceeds the input size of the model. Specifically, we show the results for model 07VEJJ (N=100 input) where clusters have 150 or 200 points per cluster (see Supplementary Figure 4).”

While the first authors reply confirm that N is a parameter that “bears major consequences”, they then indicate that the choice of N can be done by, a not very well defined, “cursory examination of the input data”. For this issue, the reviewer need a higher amount of details on how this examination can be done and how much this procedure is subjective and error-prone. The second author reply is instead unsatisfactory, it is hard to judge, from a single experiment (i.e. N=100) how the choice of the number of nearest neighbours affect performance. Since this parameter is so important for the approach a deeper analysis is necessary to understand its influence. On synthetic data this can be easily done since ground truth is available. At increasing values of N it is possible to evaluate Precision, Recall and F1. This test is necessary in order to reply to our previous questions. Moreover, another qualitative evaluation on real data should be presented at different values of N to grasp the real differences in changing the parameter.

Firstly, we have now added text to the manuscript detailing how to determine an appropriate value for the number of nearest neighbours. CAML is intended for the assessment of biological data and therefore anticipates a certain type of point-clustering, for example: points (and clusters) are constrained to two or three spatial dimensions and that there is a realistic limit on the density of points that may be feasibly (physiologically) contained within a cluster. We demonstrate in the manuscript analysis allowing clusters exceeding 70,000 points per μm^2 (e.g. for 25 nm clusters with 150 points), many times what is typically observed in biological data. We have now also added the evaluation of N which the Reviewer requests, evaluating Precision, Recall and F1 as a function of N (Supplementary Figure 9) for a variety of new simulated datasets. In addition, we have performed the same evaluation with Experimental data of the protein LAT in T cells (Supplementary Figure 10). The analysis works as expected, with performance scores dropping gradually if the number of points per cluster exceeds a model's input size N.

The next major aspect that requires clarification and balancing in the paper is still related to the dataset bias. It is a fact in the machine learning community that any synthetic dataset brings implicitly a bias, which is related to the different statistics between real and generated data. Sentences in the paper like:

“Although the models described here were trained on simulated clusters with hard edges and circular shapes, they are nevertheless able to find clusters of abstract and arbitrary shapes, including highly elongated clusters”

are strong statement even if supported by some experimental evidence (see later). Even more because it is clear that the method is synthetic data dependent as hinted by the very same authors when saying:

"For example, our models are not appropriate to recover clusters containing more points than the models were trained to check"

Here there are then two requests. Balance the paper in its conclusions, there will always be a gap between real and synthetic data unless different methods are used (e.g. GAN). Be also precise over the capability of the network of generalising on elongated and other clusters. The paper mention a “A qualitative assessment” when evaluating elongated structures but the question is why was not possible to do a quantitative evaluation to measure the recall, precision and F1 over the recovered elongated clusters. It is very useful to understand which is the drop of performance in using clusters of different shape (it has to be a drop, validation on them possibly is not performed) and analyse in depth this data.

We thank the Reviewer for their comments and have balanced the conclusion by stating that simulated data can never completely match experimental and therefore has the potential for bias. We have also included additional analysis with fibrous (elongated) data with ground-truth information (Supplementary Figure 14), where we have included a confusion matrix to indicate the model's performance on these data.

As a general comment, the conclusion of the paper should state clearly the shortcomings to have a balance judgement of this work related to:

- Synthetic dataset bias **Added statement, as noted above.**
- Drop in performance when considering different cluster arrangements **We have added discussion as well as the data on performance, as detailed above**
- The parameter N has an influence on results **We have added discussion as well as the data on performance, as detailed above**

- Euclidean norm is a design choice (with its own properties) but it might not be the best metric with other cluster rearrangements (especially in 3D) **Again, we have added to the discussion on the choice of Euclidean geometry, as requested.**

REVIEWERS' COMMENTS:

Reviewer #1 (Remarks to the Author):

The authors answered my concerns. I do recommend this work for publication.

Reviewer #2 (Remarks to the Author):

The authors have provided very convincing arguments and greatly balanced the conclusions of the approach. This would help researchers to focus on the possible shortcomings of learning based clustering approaches. Before acceptance there are very minor points to consider in the paper that can be easily fixed.

Minor comment, Supplementary Figure 8, line 793: I would not include timing for loading data and exporting data since this is not related to the methodology but to a data parsing method (i.e. it is not of interest for the method analysis).

Line 267: There is a reference error as the text appears as: "Supplementary Figure 10Error! Reference source not found."

Line 467: "Pointllist", not clear what the term refers to

Reviewer 2 Comments

- *Minor comment, Supplementary Figure 8, line 793: I would not include timing for loading data and exporting data since this is not related to the methodology but to a data parsing method (i.e. it is not of interest for the method analysis).*
 - We have added a new plot to Supplementary Figure 8 (as panel (b)) featuring the breakdown of timings for different processing functions (including one for file I/O time, which is minimal) used by our method.
 - We feel that the overall time taken to perform different clustering methods is of more interest to readers than the time taken for any specific single computational operation and so have retained the original timings plot (as Supp Fig 8a). Furthermore, as the comparisons were performed on the same machine using the same data, any time taken by 'data parsing' ought to be roughly equivalent for each method.
- *Line 267: There is a reference error as the text appears as: "Supplementary Figure 10Error! Reference source not found."*
 - We do not see this error in our copy. In any case, we have removed and reinserted the figure reference field.
- *Line 467: "Pointillist", not clear what the term refers to*
 - We have changed both occurrences of 'pointillist' to 'point pattern'.